# The Halotolerant Rhizobacterium—*Pseudomonas koreensis* MU2 Enhances Inorganic Silicon and Phosphorus Use Efficiency and Augments Salt Stress Tolerance in Soybean (*Glycine max* L.)

**DOI:** 10.3390/microorganisms8091256

**Published:** 2020-08-19

**Authors:** Arjun Adhikari, Muhammad Aaqil Khan, Ko-Eun Lee, Sang-Mo Kang, Sanjeev Kumar Dhungana, Narayan Bhusal, In-Jung Lee

**Affiliations:** 1School of Applied Biosciences, Kyungpook National University, Daegu 41566, Korea; farmerarjun7@gmail.com (A.A.); aqil_bacha@yahoo.com (M.A.K.); ge8340@daum.net (K.-E.L.); kmoya@hanmail.net (S.-M.K.); 2Department of Southern Area Crop Science, National Institute of Crop Science, Rural Development Administration, Miryang 50424, Korea; sanjeev@korea.kr; 3Department of Forest Sciences, Seoul National University, Seoul 08826, Korea; bhusal.narayan4@gmail.com

**Keywords:** silicon, phosphorus, solubilization, *P. koreensis* MU2, soybean, salinity

## Abstract

Optimizing nutrient usage in plants is vital for a sustainable yield under biotic and abiotic stresses. Since silicon and phosphorus are considered key elements for plant growth, this study assessed the efficient supplementation strategy of silicon and phosphorus in soybean plants under salt stress through inoculation using the rhizospheric strain—*Pseudomonas koreensis* MU2. The screening analysis of MU2 showed its high salt-tolerant potential, which solubilizes both silicate and phosphate. The isolate, MU2 produced gibberellic acid (GA_1_, GA_3_) and organic acids (malic acid, citric acid, acetic acid, and tartaric acid) in pure culture under both normal and salt-stressed conditions. The combined application of MU2, silicon, and phosphorus significantly improved silicon and phosphorus uptake, reduced Na^+^ ion influx by 70%, and enhanced K^+^ uptake by 46% in the shoots of soybean plants grown under salt-stress conditions. MU2 inoculation upregulated the salt-resistant genes *GmST1*, *GmSALT3*, and *GmAKT2*, which significantly reduced the endogenous hormones abscisic acid and jasmonic acid while, it enhanced the salicylic acid content of soybean. In addition, MU2 inoculation strengthened the host’s antioxidant system through the reduction of lipid peroxidation and proline while, it enhanced the reduced glutathione content. Moreover, MU2 inoculation promoted root and shoot length, plant biomass, and the chlorophyll content of soybean plants. These findings suggest that MU2 could be a potential biofertilizer catalyst for the amplification of the use efficiency of silicon and phosphorus fertilizers to mitigate salt stress.

## 1. Introduction

Exposures to salt stress are a major threat to sustainable agricultural production and food security [1]. The world population is estimated to reach 9 billion by 2050 and approximately 800 million hectares of cultivable lands in the earth have high salt concentration [2]. Salt stress has adverse effects on plant metabolism, which include osmotic stress, ionic imbalance, oxidative damage, reduced water acquisition, damage of cellular structures, and declination in gas exchange rates, which ultimately leads to plant death [3,4].

Plants acquire several strategies to tolerate salt stress, and ionic adjustment is the major one. Here, plants avoid the influx of Na^+^ and Cl^−^, which causes ionic toxicity and accumulation of K^+^ [5]. This adverse effect on the ions-utilizing system alters the biochemical and metabolic processes of plants when the intake of Na^+^ exceeds its efflux through the plasma membrane channels [6]. The ionic system in plants has been reported to be extensively linked to the ideal concentration of phosphorus and silicon [7] Bargaz et al. [8] reported that phosphorus (P) supplies induced salt tolerance in *Phaseolus vulgaris* through acquisition of K^+^ and Ca^++^. Similarly, several literature studies have reported the role of silicon (Si) in improving plant physiology, growth, and yield [9]. The enhancement of P and Si uptake in plants has reduced several stresses of plants such as biotic (pathogens) [10] abiotic (heavy metals) [11], water [12], thermal [13], and salt stresses [14]. Hence, proper utilization of Si and P can improve plant metabolism and growth under diverse environmental conditions.

Several efforts have been made to improve the utilization efficiency of inorganic fertilizers, however less success has been achieved [15]. Researchers have warned that long term continuous application of inorganic fertilizer without adequate organic restitution may degrade the soil to the level where it may become extremely unresponsive or even non-responsive to inorganic fertilizer [15,16,17]. The earth crust comprises approximately 28% of silicon [18]. However, nutrients are supplied through external fertilizer, due to its low bioavailability for plant uptake. After Ma and Yamaji first identified the Si transport mechanism through silicon transporter genes LSi1, LSi2, and their homologous genes in rice, silicon has been recognized as a beneficial element [19,20]. Similarly, P has been considered as a major element for the entire commercial crop throughout the world. According to the FAO [21], the world consumption of nitrogen, phosphorus, and potash (NPK) was approximately 186.67 million tons in 2016, in regard to this P demand is forecasted to increase by 2.2% in 2020, but the utilization efficiency by plants is limited to 10–15% [22]. This has resulted in low productivity, high production cost, and has adversely affected poor farmers globally [23]. In addition, the incremental growth of high P application has imposed several environmental threats such as water eutrophication, heavy metal accumulation, and phytotoxicity [22]. Hence, a judicious approach is required regarding fertilizer application to achieve optimum benefits without adversely affecting the environment.

The plant growth promoting bacteria (PGPB) is known to be an emerging biofertilizer, which has been widely reported as a biologically safe tool to enhance crop productivity [23]. The phosphate-solubilizing ability of microorganism is considered to be one of the major functional aspects of PGPB and has been cited in several literature studies [24,25,26,27]. It has been reported that several phosphate-solubilizing PGPBs promote plant growth under salt stress such as *Rhizobium* and *Pseudomonas* (maize) [28], *Acinetobacter* sp. and *Bacillus* sp. (*Phyllanthus amarus*) [29], *Enterobacter* sp. (wheat), but the influence of silicate solubilizing bacteria in plants under stress has been ignored. The microbial interaction regulates plant physiology through the production of endogenous phytohormones, signal transmission, secondary metabolite production, increasing nutrient bioavailability through metal dissolution, and improving absorption [30]. Under stress, they augment plant physiology through the modulation of phytohormones such as abscisic acid (ABA), jasmonic acid (JA), and salicylic acid (SA) [31]. In addition, it has been reported that PGPBs strengthen antioxidant systems through the regulation of amino acid under stress conditions [32]. Moreover, it has been widely reported that PGPB plays a functional role in reducing Na^+^ uptake and enhancing K^+^ ion uptake in plants, which is the prime strategy for osmotic adjustment during salt stress [33]. However, Etesami [34] doubts the beneficial impact of PGPBs when applied alone and highlighted the significance on the combined use of Si and PGPBs as a potent combination to help plants tolerate biotic and abiotic stresses.

Salinity has been a major limiting factor for the growth of several leguminous plants such as soybean [35]. Since globally, soybean is known as a highly nutritive plant used for edible oil production due to the high protein and oil content of its seeds [36], the production of this crop is at higher risk due to salt stress [37]. In addition, massive fertilizer application is an emerging threat to the environment. Although Si, P, and microorganisms plays a key role under several biotic and abiotic stresses, there is a lack of information regarding microorganisms with plant growth promoting traits that could efficiently enable Si and P uptake and hence mitigate salt stress in plants. In this study, we identified a salt-tolerant plant growth promoting rhizobacteria and assessed its silicate and phosphate solubilization ability. Here we introduced the effect of halotolerant (silicate and phosphate solubilizing) rhizobacterium that would not only enhance Si and P uptake but also mitigate salt stress through the modulation of the internal physiology of soybean plants.

## 2. Materials and Methods

### 2.1. Microbe Selection

Several rhizospheric bacteria from diverse locations of the Daegu agricultural fields were collected and stocked in the crop physiology department of Kyungpook National University where they were screened for silicate and phosphate solubilization. The strains were cultured using Petri plates in the national botanical research institute’s phosphate medium containing 0.25% [Ca_3_(PO_4_)_2_] to detect phosphate solubilization and in glucose agar medium (10 g L^−1^ of glucose) containing 0.25% (Mg_2_O_8_S_13_) to detect silicate solubilization after which they were cultured at 28 °C for 10 d. Both media were supplemented with different salt concentrations ranging from 100–400 mM. The bacteria with high Si and P solubilizing ability under salt stress were selected for further experiments (Figure 1). They were identified through 16 s rRNA gene sequence analysis as previously reported in the National Center for Biotechnology Information (NCBI) database with accession number KP676116 Appendix B.

### 2.2. Quantification of Gibberellins and Organic Acid

An amount of 200 mL of bacterial culture in Luria–Bertani media was grown under normal and salt-stressed conditions (200 mM) for a week. Bacterial culture (50 mL) was used for each replication and gibberellic acid (GA) was extracted according to the method described by Lee et al. [38]. Using a gas chromatograph coupled to mass spectroscopy (GC-MS, 6890 N network GC system, and 5973 network mass selective detector; Agilent Technologies, Santa Clara, CA, USA) the extracted GAs was quantified.

However, organic acid was detected by the method described by Lee et al. [39]. Quantification was conducted by high-performance liquid chromatography (HPLC; Waters 600, Milford, MA, USA) using a PL Hi-Plex H column (7.7 × 300 mm, Waters Co., Milford, MA, USA), detector refractive index (Waters 410, Milford, MA, USA), and 5 mM H_2_SO_4_ as the solvent in distilled water The injection volume was 20 µL, with oven temperature of 65 °C and flow rate of 0.6 mL min^−1^.

### 2.3. Plant Materials and Experiment

#### 2.3.1. Screening of Si and P Dose

Equal sized seedlings of soybean var. pungsanmul were treated using randomly selected doses (1–5 mM) of Si (Na_2_SiO_3_), and potassium phosphate (KH_2_PO_4_). The morphological parameters and chlorophyll contents were recorded after three weeks. Optimum growth was observed at 2 mM (Si) and 3 mM (P) (Appendix A). So, further experiments were conducted by considering 2 and 3 mM of Si and P, respectively.

#### 2.3.2. Soil Preparation

Before the experiment the soil was prepared according to the experiment design control 1 (−NaCl, without salt stress), salt (+NaCl, with salt stress), bacteria (NaCl), silicon (NaCl), silicon + Ba (NaCl), phosphorus (NaCl), phosphorus + bacteria (NaCl). Soil was treated accordingly using bacterial cell culture (optical density of 1.0) diluted with distilled water (400 mL/Kg of soil), 2 mM Si (300 mL/Kg of soil), 3 mM P (300 mL/kg of soil), and distilled water in control-1. Autoclaved soil consisting of peat moss (13–18%), perlite (11%), zeolite (6–8%), and coco-peat (63–68%), which contained the macronutrients NO_3_ (~0.205 mg/g), P_2_O_5_ (~0.35 mg/g), NH_4_^+^ (~0.09 mg/g), and K_2_O (~0.1 mg/g) were kept in a growth chamber (day/night cycle: 12 h; 28 °C/12 h; 20 °C; relative humidity (RH) 68%; light intensity 1000 μmm^−2^s Natrium lamps). The moisture level was maintained at 80–100% by the addition of distilled water, which was estimated using the moisture content of the soil, the soil pH, and humidity tester (Model DM-5, Takemura Electric Works, LTD., Tokyo, Japan).

#### 2.3.3. Plant Experiment

An amount of 350 g soil prepared as mentioned above in Section 2.3.2 was transferred in each pot and two-week old soybean seedlings were transplanted. After one week, the plants were treated using 50 mL of 150 mM saline solution per pot for 4 d. Upon completion of stress treatment, plants were irrigated using distilled water for one week after which the plants were harvested, morphological parameters were measured, and biochemical analyses were performed.

### 2.4. Chlorophyll Measurement

The chlorophyll content was measured using a Chlorophyll Content Meter (CCM-300, Opti-Sciences, Hudson, NH, USA). The emission ratio of far-red fluorescence at 735 nm and red fluorescence at 700 nm were determined.

### 2.5. Biochemical Analysis

#### 2.5.1. Quantification of Si, P, Na, and K

The mineral content of the plant shoot was determined by the method described by Bilal et al. [40]. In brief, the freeze-dried samples were suspended with HNO_3_ and digested with H_2_O_2_. The solvent obtained was quantified using ICP-MS (Optima 7900DV, PerkinElmer, Waltham, MA, USA).

#### 2.5.2. RNA Isolation and qPCR Analysis

RNA was extracted according to the method described by Chan et al. [41]. The obtained RNA was used for cDNA preparation and subjected to qPCR (Illumina, San Diego, CA, USA). Actin was used as a housekeeping gene to standardize the expression level (Table 1). 

#### 2.5.3. Quantification of Phytohormones

**ABA**-The method described by Shahzad et al. [42] was used to determine ABA. In brief, 0.5 g of lyophilized ground samples was extracted with isopropanol and acetic acid (95:5 v/v). The solution obtained was filtered, and 1 mL of the internal ABA standard ((±)-3,5,5,7,7,7-d6)-ABA) was added. The extract was washed using 1 N NaOH, and the pH was adjusted to 12–13 followed by the removal of chlorophyll through CH_2_Cl_2_. The solution was separated by ethyl acetate, concentrated, and collected by washing using a phosphate buffer (pH 8.0). Polyvinylpolypyrrolidone was added, stirred for 30 min and filtered. The filtrate was evaporated and ABA was extracted through methanol/diethylether (2:3 v/v). ABA extracts were analyzed by GC-MS/SIM (6890 N Network GC System and 5973 Network Mass Selective Detector; Agilent Technologies, Santa Clara, CA, USA).**JAs**-The method described by Adhikari et al. [43] was used to determine the JA content of the plant. In brief, 0.1 g freeze-dried ground sample was extracted using acetone/citric acid (70:30 v/v) and filtered. The (9,10–2H2) Dihydro-JA (100 ng) was added as an internal standard. The solvent was evaporated, washed using a 0.1 m potassium phosphate buffer, and the pH was adjusted to 2.5. The solution was stirred using diethyl-amino cellulose, filtered, and washed with chloroform in a column fixed with sodium sulfate. The extract was dried, washed with diethyl ether and filtered with chloroform/isopropanol (2:1), and passed through diethyl ether/acetic acid (98:2) using an amino cartridge. After filtration, the extracts were dried and recovered with diethyl ether. Methylation was performed using diethyl ether followed by dichloromethane. The peaks were obtained by GC-MS/SIM (6890 N Network GC System and 5973 Network Mass Selective Detector; Agilent Technologies, Santa Clara, CA, USA).**SA**-SA was extracted and quantified using the method by Seskar et al. [23]. In brief, 0.5 g of lyophilized samples were extracted with 100% methanol, vacuum-dried, and suspended in trichloroacetic acid (5%). The solution was partitioned by ethyl acetate, cyclopentane, and isopropanol after which it was left to dry using N_2_. The excitation and emission peaks at 305 nm and 355 nm were recorded using HPLC (Shimadzu RF-10AXL fluorescence detector).

#### 2.5.4. Antioxidant and Extent of Lipid Peroxidation Analysis

The method described by Adhikari et al. [44] was used to measure polyphenol content. In brief, samples were extracted with 100% methanol and measured using a spectrophotometer (Multiskan GO; Thermo Fisher Scientific, Waltham, MA, USA) at 750 nm. Glutathione (GSH) content was determined as described by Khan et al. [45], and the extent of lipid peroxidation was determined as described by Abd Allah et al. [46].

#### 2.5.5. Proline Content

Hydrolysis was performed for proline analysis according to the method described by Kang et al. [47]. First, 6 N HCl was added to 0.05 g of the lyophilized powdered samples and charged with N_2,_ after which they were allowed to stand for 24 h at 110 °C. The solvent was completely dried at 80 °C and washed with 0.01 N HCl. The samples were filtered and analysis was performed using an automatic amino acid analyzer (L-8900; Hitachi, Tokyo, Japan).

### 2.6. Statistical Analysis

The current study was conducted using a completely randomized design. In the plant experiment each treatment had eight replicates. All statistical analyses were performed by Duncan’s multiple range test using SAS version 9.2 (Cary, NC, USA) and GraphPad Prism software (San Diego, CA, USA).

## 3. Results

### 3.1. Quantification of Organic and Gibberellic Acids

Organic acids such as malic, citric and acetic acid were significantly increased under salt-stress conditions; however, compared with the control, no significant difference was observed in tartaric acid. GA_1_ was slightly reduced, whereas GA_3_ showed no significant difference compared to the control (Table 2 and Table 3).

### 3.2. Measurement of Morphological Parameters

Overall, the application of Si + MU2 showed an increase of 61%, 28%, 69%, and 42%, P + MU2 showed increases of 68%, 33%, 65%, and 58%, and Si + P + MU2 showed increases of 70%, 36%, 78%, and 73% in the root length, shoot length, root weight, and shoot weight compared to control 2. In comparison with the sole application of either Si or P, and a combination of both (Si + P + MU2) it increased root length, shoot length, root weight, and shoot weight by 31%, 17%, 17%, 23% and 25%, 18%, 28%, and 29%, respectively. The current results suggest that the sole application of MU2, Si, and P also improved plant growth, however, the combined application of MU2, Si, and P greatly promoted the morphological characteristics under stress conditions (Table 4).

### 3.3. Chlorophyll Measurement

Chlorophyll content is a prime indicator of photosynthetic activity. The results revealed that there was a significant drop in chlorophyll content under salt stress. Upon inoculation with MU2 culture, there was an increase in chlorophyll content; Si + P + MU2 caused an increase of 48%, Si + MU2 caused an increase of 29%, and P+MU2 caused an increase of 40% compared to control 2. The sole application of MU2, Si, and P increased chlorophyll content by 24%, 21%, and 19%, respectively compared with control 2 (Figure 2).

### 3.4. Analysis of Physiological Parameter

#### 3.4.1. Quantification of Mineral Elements (Si, P, Na, and K)

This study revealed that the combination of Si and P with MU2 significantly downregulated Na^+^ ion while, it enhanced the uptake of Si, P, and K^+^. The sole application of Si increased the silicon content of plants by 69%, by 87% with MU2, and by 121% with P + MU2 compared to control 2. Similarly, the sole application of P increased the phosphorus content of plants by 61%, by 86% with MU2, and by 106% with Si + MU2 compared to control 2. Compared to the sole application of Si or P, Si + P + MU2 increased Si uptake by 22% and P uptake by 19%. Moreover, Si + P + MU2 significantly reduced the influx of Na^+^ ions by 70% and upregulated K^+^ ions by 46% compared with control 2 (Figure 3).

#### 3.4.2. *P. koreensis* MU2 Modulates Stress-Resistant Gene in Soybean Leaves

In this study, the salt-tolerant genes *GmST1*, *GmSALT3*, and *GmAKT2* genes were highly expressed when treated with Si and P along with MU2. The expression levels in the sole application of MU2, Si, and P showed minor differences, however, a higher expression was observed when MU2 was applied with either Si or P and the maximum expressions of all three genes were recorded following Si + P + MU2 inoculation in plants (Figure 4).

#### 3.4.3. Effect on Phytohormones Level of Soybean Shoot

Phytohormone levels signify the lethal effects of stress conditions. Here the levels of endogenous ABA and JA were significantly elevated under stress. The sole application of MU2, Si, and P showed a decrease in ABA of 24%, 35%, and 23%; JA of 25%, 37%, and 24%; and an increase in SA of 43%, 15%, and 44%, respectively compared to control 2. Similarly, on Si + P + MU2-treated plants, ABA level was reduced by 47% and JA by 49% whereas, SA content increased by 85% (Figure 5).

#### 3.4.4. Effect on Plant Antioxidant System and Lipid Peroxidation under Stress Conditions

In this study, the plant showed significantly higher malondialdehyde (MDA) content under salt-stress conditions, while the combined application of MU2 with Si and P strengthened the antioxidant system through the enhancement of the reduced GSH content. Here, the Si + P + MU2-treated plants showed a 66% higher GSH content compared to the control whereas, lipid peroxidation was reduced by 50%. In GSH content, minor differences were observed in the sole application of MU2, Si, and P; however, considerably higher differences were observed in lipid peroxidation (Figure 6).

#### 3.4.5. Proline Content

Our results revealed that the inoculation of MU2 is involved in the reduction of proline content in plants under stress conditions. Here the sole application of Si and P increased the proline content by 17% and 15%, respectively, however, the sole application of MU2 reduced the proline content by 15% compared to control 2. The proline content of Si + P + MU2-treated plants reduced by 23% and 28% compared to the sole application of Si and P (Figure 7).

## 4. Discussion

Abiotic stresses such as salt stress and fertilizer pollution have been a major cause of low as well as toxic agricultural production, together with environmental hazards [48,49]. The judicious application of silicon, phosphorus, and plant growth promoting rhizobacteria is considered pivotal in the mitigation of such abiotic stresses [43]. It has been reported that Si maintains plant strength through the regulation of the internal physiology [50] and compensating P deficiencies in plants [51]. It has also been reported that PGPB engineers the root system of plants, thereby facilitating nutrient uptake [52]. Based on these scenarios, we devised an integrated approach, which included the combination of inorganic silicon and phosphorus with rhizobacteria to minimize fertilizer application and to help plants tolerate salt-stress conditions. Here, we identified plant growth promoting halotolerant rhizobacteria *P. koreensis* MU2, which has an innate potential to produce organic acid, gibberellin, and efficiently solubilize silicate and phosphate.

Higher concentration of soil salinity causes ionic imbalance and leads to osmotic stress, ion toxicity, and mineral imbalance [5]. Ion toxicity induces the efflux of cytosolic Ca^++^ and K^+^ thereby disrupting cellular homeostasis, retarding growth, and causing several physiological and morphological disorders in many crops [53]. Our results showed salt stress suppresses plant growth, biomass, and mineral uptake, however, the inoculation of MU2 with inorganic Si and P significantly improved plant growth and development under salt-stress conditions. Salt-tolerant plant growth promoting microorganisms have been reported as an alternative biofertilizer, which mitigates salt stress by producing different bioactive secondary metabolites and phytohormones including gibberellins [54]. Since, MU2 has been reported in the production of biologically active GA_1_ and GA_3_, the possible secretion of GA might have affected the enhancement of plant growth and stress mitigation. Our results are justified by the findings of several authors where GA production by microorganism promoted growth and relieved salt stress in various plants [55,56,57,58,59].

Salt stress stimulates chlorophyllase activity, degrades pigment protein, lowers chlorophyll, and has detrimental effects on plant growth, as seen in *Ephedra alata*, *C. arientinum*, and *Sesbanian sesban*. [46]. Our findings revealed that salt stress decreased chlorophyll content; however, the inoculation of isolate MU2 with Si and P mitigated salt stress and increased the chlorophyll content of soybean plants. Our results correlated with those of Baset et al. [60] and Bal et al. [61] who reported a significant beneficial role of PGPR in enhancing chlorophyll content. The growth and development of plants by PGPR are fundamentally based on nutrient acquisition from roots. Moreover, the stimulation of growth and chlorophyll pigment synthesis of plants by PGPR might be due to the enhanced nutrient uptake through the root, which maintains the osmotic balance in cells thereby improving metabolism.

Excessive salt concentration hinders the uptake of essential nutrients, and PGPB greatly enhances the compartmentalization and exclusion of toxic ions through the assimilation and translocation of soil nutrient [62]. In our results, Si and P contents in plant shoots were significantly elevated upon inoculation of the MU2 isolate. Since MU2 has been previously reported in the production of organic acids and those organic acids have been widely reported for the dissolution of elements the possible effect of organic acid production by MU2 might have been to make sufficient bioavailability of these elements for plant uptake. Previously, Othman et al. [63] and Iqbal et al. [64] reported that phosphate solubilizing and organic acid producing PGPB, facilitate P and essential nutrient uptake from the soil. Our results are in agreement with Kang et al. [65] who found that the inoculation of CS4-2 enhanced the growth of Japonica rice through insoluble Si solubilization. Moreover, our results are supported by Çimrin et al. [66] who demonstrated that P application alleviated salt stress in pepper through the assimilation of N, P, K, Mg, Ca, Fe, Zn, Mn, and Cu contenta of shoot and root. The tolerance level of plants under high salt concentrations was determined with respect to the ability of plants to compartmentalize, transport, mobilize, and extrude Na^+^ ions [67]. Our results showed a higher Na content in the salt stress control plant, however a huge decline in Na^+^ ion was noted when MU2 was inoculated with Si and P in soybean plants. Consistent with the high Si, P, and K accumulation and exclusion of Na^+^ ion in soybean under salt-stressed condition in the presence of MU2, a higher expression of the salt-resistant genes *GmST1*, *GmSALT3*, and *GmAKT2* was observed. This trend of expression level could be attributed to the interaction of MU2 with soybean plants thus, producing hormones and organic acids, which help reduce osmotic stress through the mediation of the antioxidant biosynthetic pathway and osmolytes, activating stress hormones, antioxidant systems, and ion regulating genes leading to salt tolerance, thereby improving salt tolerance in soybean plants. These findings agree with Bharti et al. [68] and Khan et al. [5] who reported the beneficial role of *Dietzia natronolimnaea* and *Arthrobacter woluwensis* in mitigating salt stress by demonstrating the expression of various ion transporters and antioxidant-enzymes regulating genes.

Understanding the interplay of Si/P and Na^+^/K^+^ ions under salt-stress conditions is vital in understanding the internal physiology of plants under various stresses since they regulate endogenous plant hormones, antioxidants, and neutralize oxidative stresses [69]. Plant hormones such as ABA, JA, and SA form a channel for creating several strategies under stress conditions [70]. High ABA and JA levels signify higher stress levels in plants and a higher SA level has been demonstrated in the healing of trauma in plants [31]. In this study, MU2 inoculation significantly reduced JA and ABA content while it enhanced SA content in soybean plants. Shahzad et al. [30] and Khan et al. [71] also found that the application of GA-producing *Bacillus amyloliquefaciens* and *Bacillus cereus* significantly reduced stress hormones and promoted plant growth and this is consistent with our results. Our findings are also in agreement with Kang et al. [72], who showed that gibberellin-producing *Pseudomonas putida* lowered ABA and JA, thereby promoting growth in soybean. The generation of ABA is based particularly on stomatal conductance [50] and has a crosstalk with JA and SA under salt stress conditions [31]. Since Si, P, and PGPB have been reported to regulate stomatal conductance, their synergistic effect might have reduced the stress hormones.

Membrane lipids are highly vulnerable and sensitive to ionic toxicity, hence malondialdehyde (MDA) content was observed in soybean plants under salt stress [73]. In addition, salt stress lead to the generation of reactive oxygen species (ROS), which causes oxidative stress, cellular toxicity, and damage to the cell structure in plants [74] To tolerate these stresses plants may develop a complex antioxidant defense system such as GSH accumulation [75]. Proline and polyphenol are the major indicators to determine stress levels in plants since they are involved in neutralizing ROS for protecting proteins and maintenance of the cell water balance [76]. In addition, glutathione plays a key role in the synthesis of phytochelatins and modifies cellular efflux and influx to detoxify the byproducts of lipid peroxidation [73]. In our results, MU2 inoculation enhanced GSH, regulated proline, and lowered MDA content, which indicates lower stress levels in plants. Similar results were shown by Hashem et al. [76] who reported that the inoculation of *B. subtilis* increased proline and polyphenol contents under salt-stress conditions. El-Esawi et al. [77] also showed that *Azospirillum lipoferum* increased nutrient acquisition, relived electrolyte leakage, improved phenolic content, and lowered MDA content of chickpea under saline conditions. Moreover, our results were supported by Ghoulam et al. [78] who revealed that increased proline accumulation enhanced salt tolerance in sugar beet treated with NaCl.

## 5. Conclusions

Overall, the plant growth promoting rhizobacterium *P. koreensis* MU2 significantly enhanced Si, P, and K uptake and promoted soybean growth under salt stress conditions by activating hormones, antioxidants, and salt-resistant genes. We hope that the current findings will open up new insights into the development of an integrated approach for the optimum utilization of chemical fertilizers to enhance crop production not only under salt stress but also for other pathogenic and environmental stresses. Based on the current evidence, an extensive study on diverse crops, different fertilizers, and stress conditions is under operation.

## Figures and Tables

**Figure 1 microorganisms-08-01256-f001:**
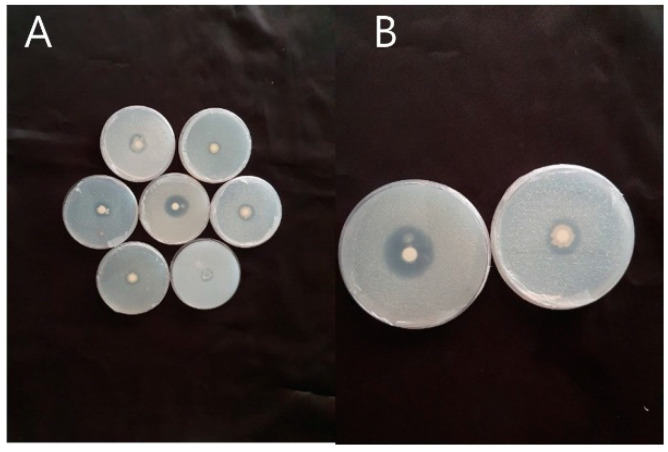
Detection of *P. koreensis* MU2 silicate and phosphate-solubilizing ability under salt-stress conditions. (**A**) Overall screening under different salt concentrations (**B**) presence of a halo-zone representing silicate and phosphate solubilization at a salt concentration of 300 mM.

**Figure 2 microorganisms-08-01256-f002:**
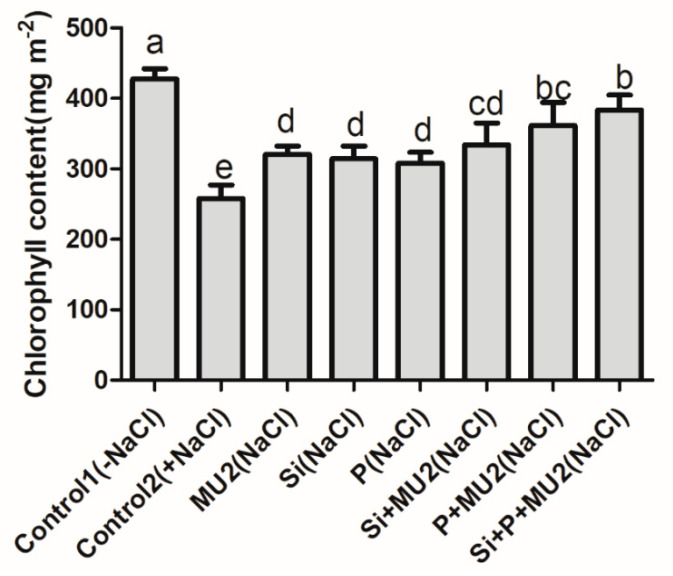
Effect of *P. koreensis* MU2 inoculation on the chlorophyll content of soybean plants grown under salt-stress conditions. Each data point represents the mean of at least six replicates. Error bars represent standard deviations. Bars with different letters are significantly different from each other at *p* ≤ 0.05.

**Figure 3 microorganisms-08-01256-f003:**
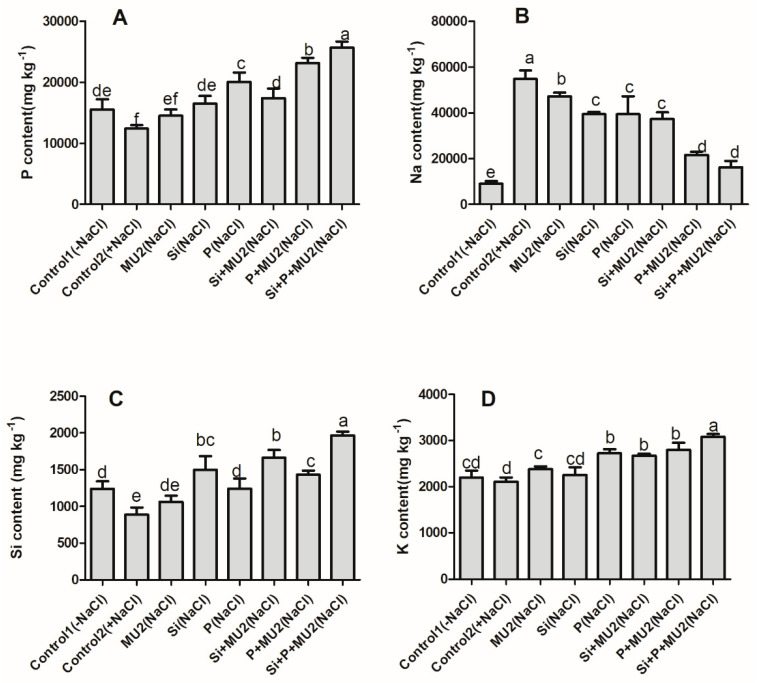
Concentration of different elements in soybean plants following the inoculation of *P. koreensis* MU2 under salt-stress conditions: (**A**) Phosphorus (P); (**B**) Sodium (Na); (**C**) Silicon (Si); (**D**) Potassium (K). Each data point represents the mean of at least three replicates. Error bars represent standard deviations. Bars with different letters are significantly different from each other at *p* ≤ 0.05.

**Figure 4 microorganisms-08-01256-f004:**
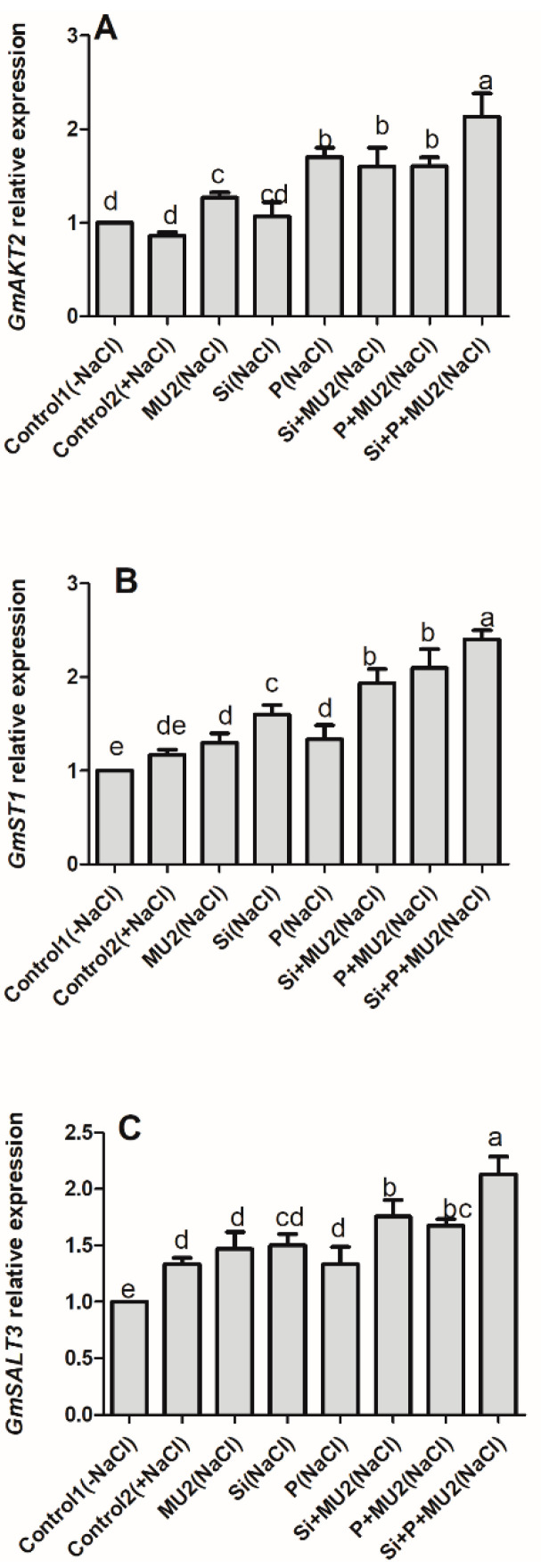
*P. koreensis* MU2 activates the transcription factor to help soybean plants tolerate salt stress. Relative expression of different genes: (**A**) K^+^ ion regulating gene *GmAKT2* salt resistant gene; (**B**) *GmST1*; (**C**) *GmSALT3*. Error bars represent standard deviations. Each data point represents the mean of at least three replications. Bars with different letters are significantly different from each other at *p* ≤ 0.05.

**Figure 5 microorganisms-08-01256-f005:**
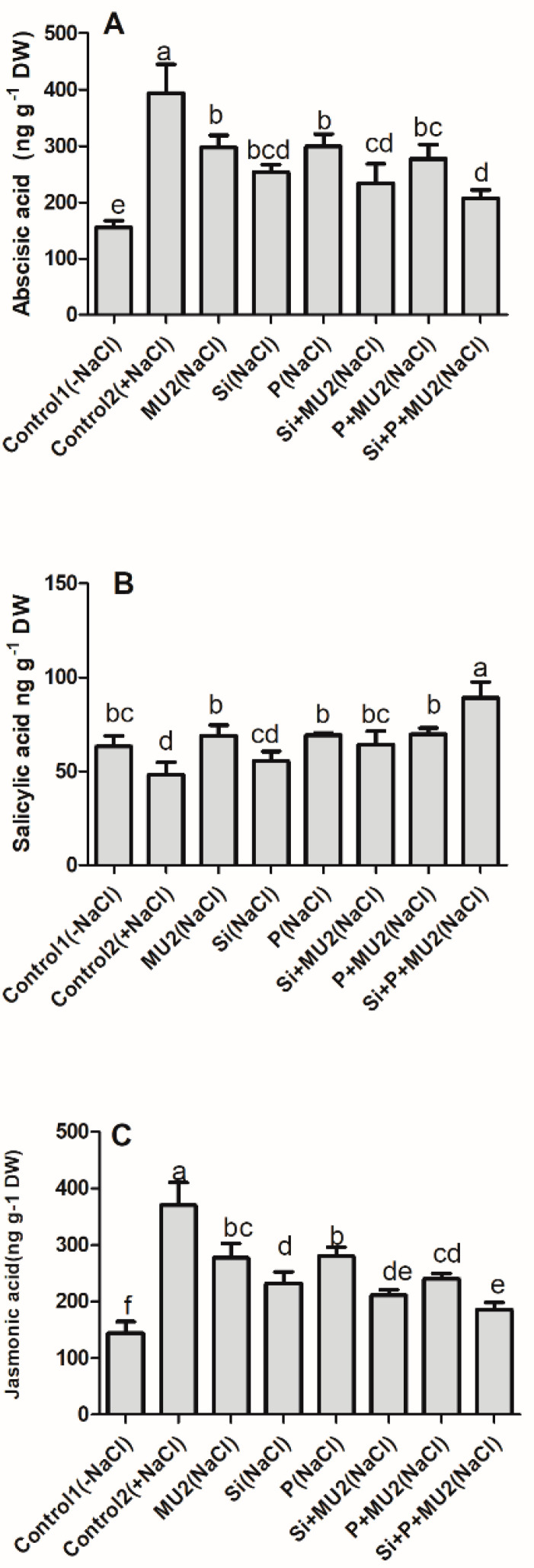
*P. koreensis MU2* regulates the endogenous phytohormones in soybean plants. Concentration of different phytohormones: (**A**) Abscisic acid; (**B**) Salicylic acid; (**C**) Jamonic acid. Error bars represent standard deviations. Each data point represents the mean of at least three replications. Bars with different letters are significantly different from each other at *p* ≤ 0.05.

**Figure 6 microorganisms-08-01256-f006:**
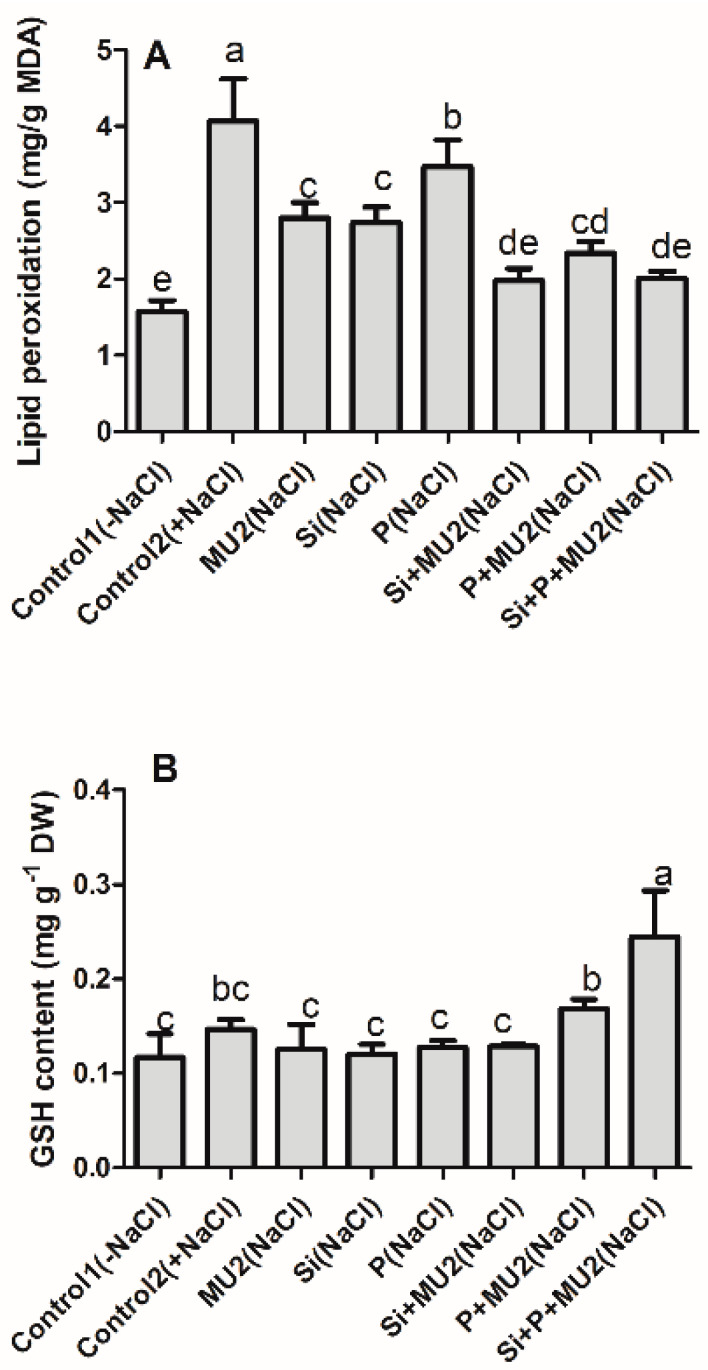
Effect of *P.*
*koreensis* MU2 inoculation in the antioxidant system of soybean plants under salt-stress conditions: (**A**) Lipid peroxidation; (**B**) Glutathione content. Error bars represent standard deviations. Each data point represents the mean of at least three replications. Bars with different letters are significantly different from each other at *p* ≤ 0.05.

**Figure 7 microorganisms-08-01256-f007:**
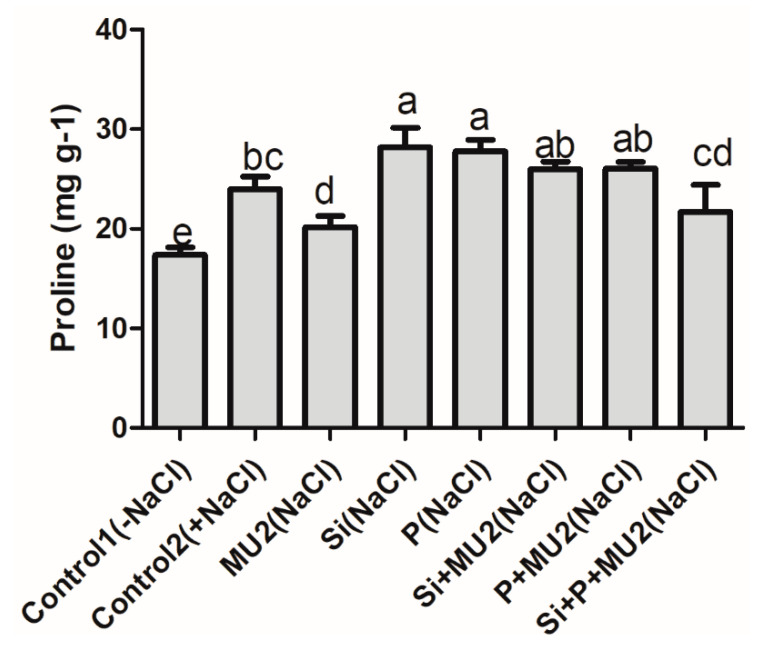
*P.**koreensis* MU2 regulates proline content in soybean plants under stress. Error bars represent standard deviations. Each data point represents the mean of at least three replications. Bars with different letters are significantly different from each other at *p* ≤ 0.05.

**Table 1 microorganisms-08-01256-t001:** Primers used for relative gene expression analysis.

Gene	Forward Primer	Reverse Primer
*GmST1*	5′TCTAGAATGGCGTTTGTTGCAGCCATG3′	5′GAGCTCTCATAAGGTTCGGGGATCCTTTC3′
*GmAKT2*	5′CACCATGCTAGCTGATCGTTACC3′	5′TCAGCTTATCCAACAAAAAAAAAGT3′
*GmSALT3*	5′ATGACGTTCAACGCGAGC3′	5′AAGTTCTTCGATAGCATCTTTA3′
*Actin*	5′ATGGTGGGTATGGGTCAAAA3′	5′GAGGACAGGATGCTCCTCA3′

**Table 2 microorganisms-08-01256-t002:** Organic acid content of *P. koreensis* MU2 pure culture under normal and salt-stressed conditions.

Organic Acid Content (ppm)	Malic Acid	Citric Acid	Tartaric Acid	Acetic Acid
Control	169.38 ± 12.36 b	200 ± 10.9 b	186.37 ± 15.88 a	95.33 ± 12.66 b
NaCl	220.63 ± 10.82 a	237.8 ± 7.12 a	219.40 ± 13.21 a	189.66 ± 21.59 a

Each value represents the mean ± SD. Each data point represents the mean of at least three replicates. Different letters in the column after mean values represent the least significant differences at *p* ≤ 0.05.

**Table 3 microorganisms-08-01256-t003:** Gibberellins acid content of *P. koreensis* MU2 pure culture under normal and salt-stressed conditions.

GA Content (ng/100 mL)	GA_1_	GA_3_
Control	6.76 ± 0.77a	11.9 ± 1.07a
NaCl	3.77 ± 0.40b	7.60 ± 0.92b

Each value represents the mean ± SD. Each data point represents the mean of at least three replicates. Different letters in the column after mean values represent the least significant differences at *p* ≤ 0.05.

**Table 4 microorganisms-08-01256-t004:** Effects of *P. koreensis* MU2 inoculation on plant growth promoting attributes of soybean plant under salt stress.

Treatments	Root Length	Shoot Length	Root Wt.	Shoot Wt.
Control 1(-NaCl)	31 ± 1a	39.6 ± 0.62a	8.74 ± 0.36a	16.78 ± 0.61a
Control 2(+NaCl)	18 ± 2c	27.97 ± 1.59d	3.92 ± 0.79f	8.41 ± 0.7g
MU2(NaCl)	23.61 ± 1.56b	32.11 ± 1.91c	5.54 ± 0.31de	10.68 ± 0.62ef
Si(NaCl)	23.33 ± 2.08b	32.39 ± 1.22c	5.93 ± 0.23cd	11.82 ± 0.82de
P(NaCl)	23 ± 2b	31.16 ± 1.10c	5.02 ± 0.08e	10.39 ± 0.47f
Si+MU2(NaCl)	29 ± 1a	36.03 ± 1.25b	6.63 ± 0.25b	12.0 ± 0.86d
P+MU2(NaCl)	30.33 ± 2.51a	37.35 ± 1.18ab	6.5 ± 0.19bc	13.31 ± 0.31c
Si+P+MU2(NaCl)	30.66 ± 2.08a	38.17 ± 1.35ab	6.99 ± 0.2b	14.63 ± 0.98b

Each value represents the mean ± SD. Each data point represents the mean of at least six replicates. Different letters in the column after mean values represent the least significant differences at *p* ≤ 0.05.

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
