# Peer review of "The Halotolerant Rhizobacterium—Pseudomonas koreensis MU2 Enhances Inorganic Silicon and Phosphorus Use Efficiency and Augments Salt Stress Tolerance in Soybean (Glycine max L.)"

_microorganisms, 2020, doi:10.3390/microorganisms8091256_

Round 1

Reviewer 1 Report

The title and aims of the research are coherent to the scope of the Microorganisms journal. The abstract is very clearly described and comprehensive.

The philosophy and structure of the paper are perfect. The study was well planned and performed and it collects a series of pieces of information. The introduction is informative, precise, and comprised of relevant content. The literary structure of the introduction is also good. Theoretical and practical reasons for the experiments are very reasonable. This manuscript represents a good and original study with new information, presented clearly and correctly. The structure and content of the paper are understandable. The authors used very progressive methods.
The screening analysis showed the high salt-tolerant potential of MU2 that solubilizes both silicate and phosphate and produced growth stimulator, gibberellic acid. The combined application of MU2, silicon, and phosphorus significantly improved silicon and phosphorus uptake in soybean plant shoot grown under salt stress condition. Authors suggest that MU2 could be a potential bio-fertilizer catalyst for amplifying Si and P fertilizer use efficiency to mitigate salinity stress
Overall, the study is of good quality and the results are innovative. The results of the experiments provide novel data and the results and discussion section deserve a closer look.
Overall, the results are presented in a comprehensible manner, with supporting charts and graphs to illustrate the statistical analyses. Avoid writing long paragraphs, try to shorten all long paras so that confusion is avoided.
Some arguments need a simpler, clearer, and tighter presentation. In Discussion, the obtained results should be better compared with analogical researches. Authors could add new conclusions, future perspectives. That is, the body of the text has to be improved in the discussion. Please read/use the following references: https://doi.org/10.1016/j.jhazmat.2020.123256; . doi: 10.1016/j.scienta.2019.108953; https://doi.org/10.1016/j.plaphy.2020.07.022; DOI 10.1007/s11738-013-1325-7
Final comments: Manuscript is interesting, associated with actual plant science trends. In my opinion, this kind of paper should be supported for publishing. I hope, the paper will help to increase the quality of Microorganisms journal, his impact factor, and to improve his market position in the scientific field. I propose acceptation of this manuscript, with recommended minor revisions
I recommend that minor revisions are done on the article before publication. The results can promote future research.

Author Response

Reviewer 1:

The title and aims of the research are coherent to the scope of the Microorganisms journal. The abstract is very clearly described and comprehensive. The philosophy and structure of the paper are perfect. The study was well planned and performed and it collects a series of pieces of information. The introduction is informative, precise, and comprised of relevant content. The literary structure of the introduction is also good. Theoretical and practical reasons for the experiments are very reasonable. This manuscript represents a good and original study with new information, presented clearly and correctly. The structure and content of the paper are understandable. The authors used very progressive methods.
The screening analysis showed the high salt-tolerant potential of MU2 that solubilizes both silicate and phosphate and produced growth stimulator, gibberellic acid. The combined application of MU2, silicon, and phosphorus significantly improved silicon and phosphorus uptake in soybean plant shoot grown under salt stress condition. Authors suggest that MU2 could be a potential bio-fertilizer catalyst for amplifying Si and P fertilizer use efficiency to mitigate salinity stress
Overall, the study is of good quality and the results are innovative. The results of the experiments provide novel data and the results and discussion section deserve a closer look. 
Overall, the results are presented in a comprehensible manner, with supporting charts and graphs to illustrate the statistical analyses. Avoid writing long paragraphs, try to shorten all long paras so that confusion is avoided.
Some arguments need a simpler, clearer, and tighter presentation. In Discussion, the obtained results should be better compared with analogical researches. Authors could add new conclusions, future perspectives. That is, the body of the text has to be improved in the discussion. Please read/use the following references: https://doi.org/10.1016/j.jhazmat.2020.123256; . doi: 10.1016/j.scienta.2019.108953; https://doi.org/10.1016/j.plaphy.2020.07.022; DOI 10.1007/s11738-013-1325-7
Final comments: Manuscript is interesting, associated with actual plant science trends. In my opinion, this kind of paper should be supported for publishing. I hope, the paper will help to increase the quality of Microorganisms journal, his impact factor, and to improve his market position in the scientific field. I propose acceptation of this manuscript, with recommended minor revisions
I recommend that minor revisions are done on the article before publication. The results can promote future research

Response: Thank you for your appreciation and suggestion. We have included the information as suggested by the reviewer with a closure look so as to maintain the fluidity of content. The entire manuscript has been edited by professional English language editor therefore, some line which reviewer has suggested might be modified.

Reviewer 2 Report

This paper has presented valuable information on the use of bacterial resources to ameliorate the salt stress and improve the solubilization of silicate and phosphate. The study has focused on the use of halotolerant Rhizobcateria to reduce the salinity stress in the soyabean plants and increased the uptake of Si and P. The overall study is appropriate and could be considered for publication. However, there are some errors prevailed and some queries existed, which the author should addressed prior considering for final acceptance. Following suggestion and comments could help to improve the manuscript:

Comments:

  1. Please see the attached file for line-by-line suggestions. These suggestion might be helpful to revise the manuscript.
  2. Add some numerical data in the Abstract.
  3. As this manuscript utilizes potent bacteria, it fails to include some literatures in the introduction regarding the information on different type of bacteria which are in application as PGP bacteria. Please include some example of bacteria from previous studies which are regarded as potent PGP. Also emphasize on silicates and phosphate solubilizing bacteria.
  4. Section 2.1 Microbe Selection: How was the bacteria collected? Do the authors themselves isolated bacteria used in this study? The author should improve on the aspect of isolation and cultivation of bacterial strains. Especially, focus on soil collection, isolation technique, and identification.
  5. In methodology, include a brief section describing the statistical analysis used in this study.
  6. Discussion section should be improved focusing on the main findings of this study.
  7. Conclusion should be short and concise and should be written based on the main objectives designed for this study.
  8. Overall English language should be improved. It is suggested to check the manuscript with the English language expert.

Author Response

Reviewer 2:

This paper has presented valuable information on the use of bacterial resources to ameliorate the salt stress and improve the solubilization of silicate and phosphate. The study has focused on the use of halotolerant Rhizobacteria to reduce the salinity stress in the soybean plants and increased the uptake of Si and P. The overall study is appropriate and could be considered for publication. However, there are some errors prevailed and some queries existed, which the author should addressed prior considering for final acceptance. Following suggestion and comments could help to improve the manuscript:

#1. Please see the attached file for line-by-line suggestions. These suggestions might be helpful to revise the manuscript.

Response: We have considered all the comments suggest on the file. We have edited the entire manuscript through professional English language editor.

#2. Add some numerical data in the Abstract.

Response: The quantity of Na+ and K+ ion in plant is considered crucial for nutrient metabolism especially under salt stress. Due to the word limit, we have decided to include the numerical data of only Na+ and K+ in the abstract which we think would reflect the core findings of our experiment.

#3. As this manuscript utilizes potent bacteria, it fails to include some literatures in the introduction regarding the information on different type of bacteria which are in application as PGP bacteria. Please include some example of bacteria from previous studies which are regarded as potent PGP. Also emphasize on silicates and phosphate solubilizing bacteria.

Response: We agree to the reviewer comment. We have included an information with regard to the PGPB in terms of phosphate and silicate solubilization under salt stress in introduction section. 

#4 Section 2.1 Microbe Selection: How was the bacteria collected? Do the authors themselves isolated bacteria used in this study? The author should improve on the aspect of isolation and cultivation of bacterial strains. Especially, focus on soil collection, isolation technique, and identification.

Response: The rhizobacterium used in this experiment was previously stored in the crop physiology department of Kyungpook National University, Daegu, South Korea which is isolated from the rice field soil of Daegu location. Several such rhizobacterium were screened for detecting the plant growth promoting potentiality from the stock of the laboratory and finally the strain with high silicate and phosphate solubilizing ability was selected. To address the question raised by reviewer we have shifted some content form results section to the materials and method sections and added some information which seems reliable to understand with regard to the isolation, selection and identification of microbes.

#5. In methodology, include a brief section describing the statistical analysis used in this study.

Response: Thank you for your suggestion. The statistical analysis section is added.

#6. Discussion section should be improved focusing on the main findings of this study.

Response: The discussion section is revised and improved accordingly. Also new statements have been added as suggested by reviewer 1.

#7. Conclusion should be short and concise and should be written based on the main objectives designed for this study.

Response: Some lines are deleted and the statements are paraphrased to minimize the conclusion to make it more concise and comprehensible.

#8. Overall English language should be improved. It is suggested to check the manuscript with the English language expert.

Response: The entire manuscript has been edited by the English language expert. The certificate of English language edition has been attached along with the submission of revised version of the manuscript.